# Prototype memory and attention mechanisms for few shot image generation

**Tianqin Li[†], Zijie Li[†], Andrew Luo, Harold Rockwell, Amir Barati Farimani, Tai Sing Lee**
Carnegie Mellon University
{tianqinl, zijiel, afluo, hrockwel, afariman, taislee}@andrew.cmu.edu

## Abstract

Recent discoveries indicate that the neural codes in the superficial layers of the primary visual cortex (V1) of macaque monkeys are complex, diverse and super-sparse. This leads us to ponder the computational advantages and functional role of these "grandmother cells." Here, we propose that such cells can serve as prototype memory priors that bias and shape the distributed feature processing during the image generation process in the brain. These memory prototypes are learned by momentum online clustering and are utilized through a memory-based attention operation. Integrating this mechanism, we propose *Memory Concept Attention* (**MoCA**) to improve few shot image generation quality. We show that having a prototype memory with attention mechanisms can improve image synthesis quality, learn interpretable visual concept clusters, and improve the robustness of the model. Our results demonstrate the feasibility of the idea that these super-sparse complex feature detectors can serve as prototype memory priors for modulating the image synthesis processes in the visual system.

## 1 Introduction

Recent neurophysiological findings based on calcium imaging have revealed that many neurons in the superficial layers of V1 are strongly tuned to complex local patterns, rather than simple oriented edges or bars. These complex neurons exhibit far stronger responses (a 3-5 fold increase) when exposed to their preferred patterns than when exposed to simple grating and bar-like stimuli. The high degree of specificity in these neurons' selectivity suggests that they might serve as specific pattern detectors (Tang et al., 2018a). Due to the selectivity towards complex stimuli, the population response of these V1 neurons is extremely sparse. Only 4–6 out of roughly 1000 neurons respond strongly to any given pattern or natural stimulus (Tang et al., 2018b). This finding is reminiscent of an earlier study that found similarly sparse encoding of concepts in the hippocampus (Quiroga et al., 2005). Recent data in V4 also indicate the existence of cells with a similarly high degree of sparsity[*]. Thus, we theorize such cells should exist in every level of the hierarchical visual system. We called these highly selective sparsely-responding feature detectors "grandmother neurons" to highlight their possible explicit encoding of specific prototypes, even though in reality, a prototype is likely represented by a sparse cluster of neurons rather than a single cell. Neurons in different layers of each visual area exhibit different degrees of response sparsity, complementing one another in various functions. The observations of this diverse set of super-sparsely responding feature detectors raise the following question: what are the possible computational benefits and rationales of having such neurons in the early visual cortex?

In this paper, we hypothesize that these "grandmother neurons" can serve as a prototype memory prior to modulate the process of image synthesis. Image synthesis is a central theme in a number of hierarchical models of the visual system, including interactive activation and predictive coding (McClelland & Rumelhart, 2020; Grossberg, 1987; Mumford, 1992; Rao & Ballard, 1999; Lee & Mumford, 2003), and is hypothesized to take place through the top-down feedback connections between areas in the visual cortex. These priors allow the synthesis process to look beyond the current spatial context and utilize the prototype memories learned and accumulated over time. Thus, the

---

[†]Equal contribution.
[*]Unfortunately the data is not publicly available yet.

"grandmother cells" serve as structural conceptual priors in a memory attention process during image generation.

To emphasize the importance of having an attention mechanism that operated beyond the current image representation, we name our proposed memory-based attentional process *Memory Concept Attention* (**MoCA**). MoCA is a module that can be plugged into any layer of pre-existing generator architectures in the GAN framework. We test our model by performing extensive experiments using the state-of-the-art StyleGAN2 (Karras et al., 2020b) and a newly proposed few-shot image generator FastGAN (Liu et al., 2021). Our experiments demonstrate the utilization of prototype information accumulated within the semantic clusters during training can improve few-shot image generation on Animal-Face Dog (Si & Zhu, 2012), 100-Shot-Obama (Zhao et al., 2020), ImageNet-100 (Russakovsky et al., 2015), COCO-300 (Lin et al., 2014), CIFAR10 (Krizhevsky et al., 2009) and Caltech-UCSD Birds (CUB) (Welinder et al., 2010). Additionally, we also find that generators with MoCA can resist a certain degree of injected noise corruption during testing time, suggesting that attending to a structured memory prior during generation can improve the robustness of the model. Our goal here is to explore the utility of prototype cells as memory priors in a standard computer vision image generation task, with a view to gaining some insights into the advantages of having these "grandmother" neurons in the visual cortex at a functional level.

## 2 RELATED WORK

**Visual Concept Learning**   Visual concepts, defined as intermediate level semantic features, have been shown to be effective in overcoming misclassification of objects due to occlusion when used in a feedforward voting scheme (Wang et al., 2017). Explicit representations of visual concepts using prototype neurons might also serve as effective reconfigurable parts for building compositional machines (Bienenstock et al., 1997; Geman et al., 2002; Zhu & Mumford, 2007). In this paper, we explore the use of visual concepts in the form of prototype memory priors to provide temporal spatial and temporal contextual modulation with an attention mechanism for the complex compositional task of image generation.

**Self-Attention** The attention mechanism  (Vaswani et al., 2017) in deep learning is popular in Natural Language Processing (NLP) and is also known as "non-local networks" in the vision community (Wang et al., 2018). Zhang et al. (2019a) introduced self-attention into generative models in the Self-Attention GAN. Since then, adding self-attention into Generative Adversarial Networks has become a standard practice. Brock et al. (2018) and Esser et al. (2020) demonstrate the benefits of using self-attention in high-fidelity image synthesis models. However, current self-attention mechanisms in GANs only utilize contextual information within the same image to modulate the activation. In this work, we propose the use of a memory cache of intermediate-level visual conceptual prototypes to provide additional modulation in addition to the standard self-attention. In our work, the activation mechanism attends not only to its spatial context within the image itself, but also attends to a cached set of memory prototypes accumulated over time.

**Prototype Memory Mechanism** A memory bank can capture diverse features over a long time period, and has been shown to be effective in other domains. Wu et al. (2018) utilizes a memory bank in contrastive learning to obtain more diverse negative samples to contrast with. Caron et al. (2020) also uses a memory queue during training to accumulate representative negative samples. He et al. (2020) shows that the stability of the features accumulated in the memory bank can be improved using a momentum-updated encoder, a strategy we will also use in learning our prototypes. SimGAN (Shrivastava et al., 2017) introduces the image pool trick which uses a buffer to store previously generated samples, in order to make the discriminator focus not only on the current training batch, but also improve itself based on memory. The major innovation of our work is the proposal, inspired by recent neurophysiological findings, that we should have memory banks at intermediate levels, and, in principle, every level of the visual hierarchy. While early memory bank work stored images at the instance level for object recognition, which is not useful for image generation, we argue that memory banks at lower levels of the hierarchy can store prototypes of parts and sub-parts which are useful for image generation. Few-shot image generation, in particular, can benefit from memory mechanisms that support flexible composition and decomposition of parts, particularly when data are limited.

**Few-Shot Prototypes Learning** Few-shot learning refers to performing computer vision tasks when training data are very limited. One of the popular ideas in the few-shot learning literature is to form distinct prototypes from the training set and use them during testing (Snell et al., 2017). Although

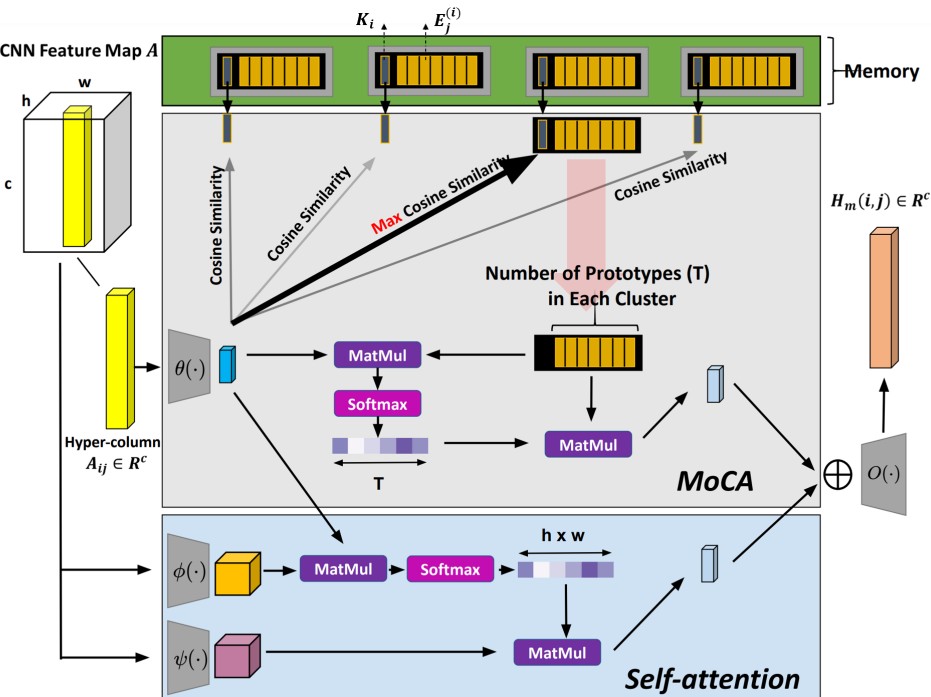

Figure 1: Attention layer using MoCA and Self-Attention. In MoCA, the input activation $A_{ij}$ is first transformed into low dimensional space via $1 \times 1$ convolution $\theta(\cdot)$ and used to select its closest semantic cell in a winner-take-all process. The selected semantic cell will allow the prototype memory cells in its cluster to participate in the MoCA process, generating a modulation that is then mapped by a 1x1 network $O(\cdot)$ from the embedding space back to the feature space. In the self-attention path, the entire feature map $A$ is transformed into key and value via two corresponding $1 \times 1$ convolution $\phi(\cdot), \psi(\cdot)$ and then attend with query vector (encoded from $A_{ij}$) and then mapped back to the feature space. Finally, the outputs from two paths are aggregated together to form the input to the next layer. Note that decoder $O(\cdot)$ and query encoder $\theta(\cdot)$ are shared across two paths.

MoCA also forms prototypes during the training stage and uses them during inference, there are two important distinctions between our work and Snell et al. (2017): (1) Snell et al. (2017) forms prototypes at the instance level whereas MoCA's prototypes are generated at the intermediate parts level. (2) Snell et al. (2017) simply selects the closest prototype to obtain discrete class prediction, MoCA, on the other hand, employs an attention process to continuously modulate the activation features based on the prototypes, hence can be applied to a broader set of tasks, including image synthesis, which predicts continuous pixel values.

**Few-Shot Image Generation** Few-shot image generation task is a challenging task as GANs are very data-hungry and inefficient. Unconditional image generation is especially difficult among the current few-shot image generation schemes. Different solutions have been developed in the literature. Recent works such as DiffAug (Zhao et al., 2020), StyleGAN-Ada (Karras et al., 2020a) propose differentiable augmentation to avoid overfitting in the discriminator. InsGen (Yang et al., 2021) proposes to use a contrastive learning objective to enhance the adversarial loss in the few-shot generation setting. These works mostly proposed methods to improve the discriminators so that they can provide better error signals to the generator, but do not specifically address the generator architecture. Another line of research proposed generator architectures to avoid mode collapse. The state-of-the-art architecture for few-shot image generation (hundreds of real images) is FastGAN (Liu et al., 2021). Here, we propose a new architectural change on the generator side and make comparison against FastGAN architecture extensively in the experiment section.

## 3 METHODS

Our key contribution is the introduction of a novel prototype-based memory modulation module to improve the generator network of a GAN. The activation from the previous layer is modified by

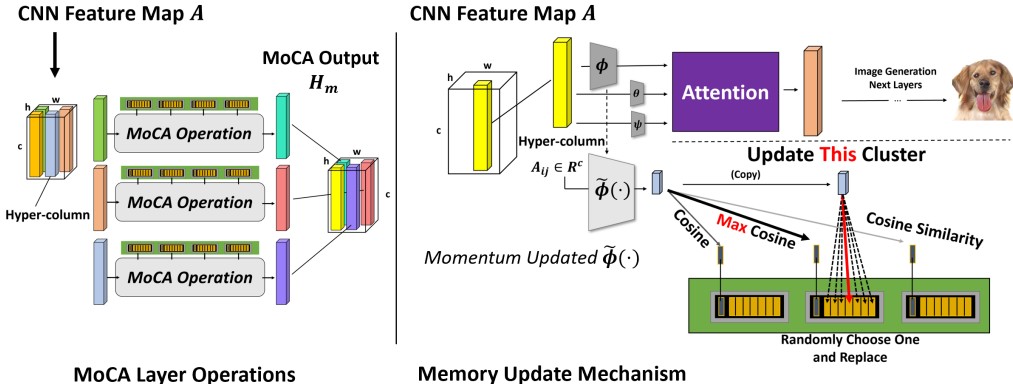

Figure 2: **Left:** MoCA Layer overview. Each hyper-column in the feature map $A$ is processed by the MoCA Operation specified in Figure 1 to generate a modulation to modify the activation of that hyper-column before passing it onto the next layer. **Right:** Memory Update Mechanism. When updating the memory, a momentum-updated projection head $\tilde{\phi}(\cdot)$ maps the hyper-column activation vector to the prototype memory space and later incorporated into the matched semantic cluster in the memory bank using a random update policy.

two attention processes: 1) contextual modulation with Memory Concept Attention (**MoCA**), and 2) spatial contextual modulation within the generated image itself (**Self-Attention**). Our module takes a feature map in the GAN hierarchy as input combines the results of these two mechanisms to modulate the feature map for further downstream processing. A high-level overview of our model is shown in Figure 1.

Formally, we denote the input to our MoCA layer to be the activation of a particular layer, $A \in R^{n \times c \times h \times w}$. The output produces a modulation $H \in R^{n \times c \times h \times w}$ that update $A$ into $\hat{A}$. To allow the information to be flexibly modulated, we first transform $A$ by $1 \times 1$ convolutions into a lower-dimensional space via functions $\theta(\cdot), \phi(\cdot)$, and $\psi(\cdot)$, where $\{\theta(A), \phi(A), \psi(A)\} \in R^{n \times \tilde{c} \times h \times w}$.

## 3.1 PROTOTYPE MEMORY LEARNING

In this section, we discuss the organization of our prototype memory that is used as one route of modulation. Our prototype concept memory is arranged hierarchically into semantic cells and prototype cells. As shown in Figure 2, each semantic cell is cluster mean representative of a cluster of prototype cells. Formally, suppose our memory $P$ consists of $M$ semantic cells $P = \{K_1, K_2, ..., K_M\}$. For each of the semantic cells $K_i$, there are $T$ of prototype cells $\{E_1^{(i)}, E_2^{(i)}, E_3^{(i)}, ..., E_T^{(i)}\}$ stored in the memory cell associated with $K_i$, where $E_j^{(i)}$ and $K_i$ both $\in R^{\tilde{c}}$ and $K_i = (\sum_{j=1}^{T} E_j^{(i)})/T$ is the mean of the stored prototype cells. These prototype cells come from the feature maps in previous iterations after being transformed via a momentum updated context encoder $\tilde{\phi}(\cdot)$ and are updated in the memory at the end of every training iteration. $\tilde{\phi}(\cdot)$ is a momentum counterpart of $\phi(\cdot)$ and its parameter is updated as shown in Equation 1 with the momentum parameter $m$. $\tilde{\phi}(\cdot)$ does not change as rapidly as $\phi(\cdot)$ so that the prototype learned are more stable, accumulating information beyond the current training batch. The rationale is similar to that in (He et al., 2020)).

$$\tilde{\phi}_\theta \leftarrow \tilde{\phi}_\theta * (1 - m) + \phi_\theta * m \qquad (1)$$

After transformed by $\tilde{\phi}(\cdot)$ to a low-dimensional space, the activation at each hypercolumn (pixel location) in the feature map is assigned to its closest semantic cluster and replaces an existing prototype cell in the memory bank of that cluster. We use a random replacement policy to prevent the prototype cells from all collapsing to trivial solutions. The update is done in a batch synchronized fashion, i.e. we update $K_i$ as the cluster mean of the $i^{th}$ cluster of prototypes after the entire memory is updated based on the recent batch.

## 3.2 MEMORY CONCEPT ATTENTION

As we described in section 3.1, our memory is a cache of hypercolumn activation patterns gathered across the entire training set as well as from different training periods. By organizing them via semantic cells ($K_i$s), we can route more diverse information into the final activation while still maintaining high data efficiency, hence yielding higher quality image synthesis task. In this section, we present the detailed operation given an existing memory $P$ and the input feature transformation $\{\theta(A), \phi(A), \psi(A)\} \in R^{n \times \tilde{c} \times h \times w}$.

During the memory attention process, an activation column $\mathbf{a} \in R^{\tilde{c}}$ from $\theta(A)$ first selects the closest semantic cell $K_i$ and retrieves the associated prototype cell matrix $\mathbf{E}^{(i)} \in R^{\tilde{c} \times H}$ from memory, where each column $j$ of $\mathbf{E}^{(i)}$ is a prototype cell $E_j^{(i)} \in R^{\tilde{c}}$ belonging to the cluster of the selected semantic cell $K_i$. The next step is to attend to the activation column $\mathbf{a}$ with the prototype cell matrix $\mathbf{E}^{(i)}$. First, a similarity score $\mathbf{s}$ is calculated as $\mathbf{s} = [\mathbf{E}^{(i)}]'\mathbf{a}$. A nonlinear softmax normalization is then applied to $\mathbf{s}$ to obtain the normalized attention weight $\boldsymbol{\beta}$ where for each entry $\boldsymbol{\beta}_t$ ($t = \{1, 2, ..., T\}$) Equation 2 applies.

$$\boldsymbol{\beta}_r = \frac{\exp(\boldsymbol{s}_r)}{\sum_{i=1}^{T} \exp(\boldsymbol{s}_l)} \tag{2}$$

Using the (softly) normalized attention weight $\boldsymbol{\beta}$, we can build the retrieved information from memory $\boldsymbol{h}_m = \mathbf{E}^{(i)}\boldsymbol{\beta}$ ($\boldsymbol{h}_m \in R^{\tilde{c}}$) for the activation column $\mathbf{a}$. By applying the same operation to the activation $\mathbf{a}$ at every spatial location as well as to every image in the batch, we obtain $\boldsymbol{H}_m \in R^{n \times \tilde{c} \times h \times w}$.

## 3.3 SPATIAL CONTEXTUAL ATTENTION

While the memory prior is important, spatial contextual information also plays a role in the activation modulation. Therefore, we also use a non-local network to implement spatial contextual modulation in the same layer. Specifically, we first compute a affinity map between $\theta(A)$ and $\phi(A)$, denoting as $\boldsymbol{S} = [\theta(A)]^T \phi(A)$. Each row of $\boldsymbol{S}$ is then normalized via softmax to allow a rather sparse attention weight $\hat{\boldsymbol{S}}$ to be computed. $\hat{\boldsymbol{S}}$ is then multiplied with $\phi(A)$ to get the spatial contextual modulation tensor $\boldsymbol{H}_s \in R^{n \times \tilde{c} \times h \times w}$.

## 3.4 INTEGRATE TWO ROUTES OF MODULATION

Finally, we integrate the retrieved information from memory $\boldsymbol{H}_m$ and the spatial contextual modulation $\boldsymbol{H}_s$ by element-wise addition, $\boldsymbol{H} = \boldsymbol{H}_m \oplus \boldsymbol{H}_s$, which is then transformed back to the original feature space via the $1 \times 1$ convolution $O(\cdot)$. A learnable parameter $\gamma$ is multiplied to the multiplication as weight and add back to the input activation, i.e. $\hat{A} = \gamma O(\boldsymbol{H}) + A$, which is then output into the next layer in the generator.

## 4 EXPERIMENTS

The proposed MoCA module is a generator architecture improvement and is not specific to the discriminator's training techniques. Note that some of the existing works on unconditional few-shot image generation (DiffAug (Zhao et al., 2020), ADA (Karras et al., 2020a), etc.) improve the quality of image synthesis by focusing on the discriminator side. For example, DiffAug proposes to perform augmentation of the generated image before being fed into the discriminator to prevent discriminator's overfitting. Although this line of work is valuable, it is *orthogonal* and *complementary* to MoCA, as MoCA is a generator architecture that can be trained under all of these discriminator training techniques. In the following experiments, we use a MoCA enhanced generator with either DiffAug or ADA augmented discriminators.

**Datasets and Metrics** We validate the performance improvements of adding the MoCA module for few-shot image synthesis with 6 datasets. Namely, we perform comparisons between the baseline architecture and the MoCA-enhanced architecture on Animal-Face Dog (Si & Zhu, 2012), 100-Shot-Obama (Zhao et al., 2020), ImageNet-100 (Russakovsky et al., 2015), COCO-300 (Lin et al., 2014), CIFAR10 (Krizhevsky et al., 2009) and Caltech-UCSD Birds (CUB) (Welinder et al., 2010). Animal-Face Dog consists of 389 dog images while 100-Shot-Obama contains 100 Obama faces with different expressions. We use 256x256 resolution for both of these datasets across all experiments. To further challenge the few-shot image generation task, we select 100 and 300 images from one class of ImageNet ("Jeep" category) and MSCOCO ("Train" category) respectively to compose

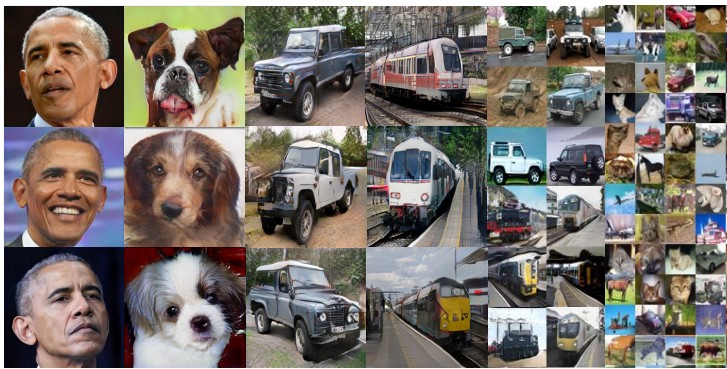

Figure 3: Generated images from MoCA on different datasets. Different resolutions of images are considered. The biggest one is 256x256, the middle one is 64x64 and the smallest one on the right is from CIFAR-10 with 32x32 resolution.

the ImageNet-100 and COCO-300 datasets (the details of the selected image list can be found in the Appendix 8.5). In addition, we also conduct experiments on CUB (5,990 256x256 resolution wild bird images) and CIFAR10 datasets (60,000 32x32 images span across 10 categories). For evaluation metrics, we use a widely adopted metric, Fréchet Inception Distance (FID) (Heusel et al., 2018) to measure the quality of generated images. For most of the experiments, except the one for the evaluation of model robustness against noise, the standard deviation of FID are often less than 1% relatively and much smaller than the performance improvement. To strengthen confidence in the evaluation of the image synthesis performance, we also evaluate the Kernel Inception Distance (KID) metric (Bińkowski et al., 2018), which is found to be more descriptive for the few-shot image generation tasks.

**Baselines** To demonstrate that MoCA is not limited to a certain backbone architecture, we add MoCA to two state-of-the-art architectures, FastGAN (Liu et al., 2021) and StyleGAN2 (Karras et al., 2020b). Of note, FastGAN is a state-of-the-art architecture that is specifically targeted on image synthesis in the *extremely* low data regime. It is an important baseline architecture that provides reasonable few-shot synthesis given only hundreds of real images and as few as 15 GPU hours of training time. Additionally, we also add MoCA to StyleGAN2, which is a more powerful and generic model but requires more data and computing resources. We will showcase that the MoCA layer can be useful for generators in both architectures in Section 4.1. The implementations of the baseline models are based on their official publicly available PyTorch implementations. We follow the best training configuration reported in the original papers to train our models.

## 4.1 Few-shot Image Synthesis Performance

Here, we present the results of the experiments on 4 small datasets, each of which has only hundreds of real images (Animal-Face Dog (Si & Zhu, 2012), 100-Shot-Obama (Zhao et al., 2020), ImageNet-100 (Russakovsky et al., 2015), and COCO-300 (Lin et al., 2014)). Qualitative results are shown in Figure 3. Meanwhile, we show quantitative comparisons between the best performance of MoCA and baseline models for each dataset in Table 1 (FastGAN) and Table 2 (StyleGAN2). Note that the KID is shown on the scale of $10^{-3}$. As shown in the results, we observe consistent improvements in terms of image synthesis quality in this difficult low data regime when adding MoCA to the generator. In terms of FID score, we observe that, using the FastGAN base architecture, MoCA can bring a 5.8% improvement on Animal Face Dog, 13.8% improvement on Obama, 21.7% improvement on ImageNet-100 and 12.4% improvement on the COCO-300 dataset (Table 1). Similar improvements are also observed with KID metrics.

To further test MoCA's enhancement on more powerful model, we run experiments with a StyleGAN2 base architecture. Using the FID evaluation metric, we observe a 5.1% improvement on Animal Face Dog dataset, 8.1% improvement on Obama dataset, 14.1% improvement on ImageNet-100 dataset and 17.3% improvement on COCO-300 dataset. Note that ImageNet-100 and COCO-300 dataset are very challenging as they contain much more diverse scenes than Animal Face Dog and Obama. To compensate for the model size of the StyleGAN2 base architecture, we downsize ImageNet-100 and COCO-300 to a 64x64 image size when training using StyleGAN2 and MoCA-StyleGAN2. From the results, we observe that MoCA-StyleGAN2 consistently synthesizes better images compared to the baseline despite the image size. Additionally, we also observe that different augmentation methods

| Generator Architecture | Discriminator Training | Animal Face Dog | | Obama | | ImageNet-100 | | COCO-300 | |
|---|---|---|---|---|---|---|---|---|---|
| | | FID ↓ | KID ↓ | FID ↓ | KID ↓ | FID ↓ | KID ↓ | FID ↓ | KID ↓ |
| FastGAN | DiffAug | 51.25 | 17.11 | 43.13 | 13.52 | 66.44 | 22.31 | 30.43 | 6.02 |
| MoCA-FastGAN (ours) | DiffAug | **48.27** | **13.83** | **37.19** | **7.81** | **52.01** | **7.23** | **26.67** | **4.53** |

Table 1: Quantitative results of MoCA on the FastGAN-based architecture. FastGAN (Liu et al., 2021) is trained following the setting in original paper, which uses differentiable augmentation (DiffAug) (Zhao et al., 2020) to train the discriminator. All experiments in this table are conducted using 256x256 size images. Results suggest a consistent improvement from the MoCA layer across different few-shot generation datasets.

| Generator Architecture | Discriminator Training | Animal Face Dog | | Obama | | ImageNet-100* | | COCO-300* | |
|---|---|---|---|---|---|---|---|---|---|
| | | FID ↓ | KID ↓ | FID ↓ | KID ↓ | FID ↓ | KID ↓ | FID ↓ | KID ↓ |
| StyleGAN2 | ADA | 58.30 | 22.72 | 28.22 | 5.44 | 54.32 | 4.53 | 77.81 | 38.66 |
| MoCA-StyleGAN2 (ours) | ADA | **55.35** | **17.62** | **25.93** | **4.91** | **46.71** | **3.03** | **64.32** | **20.53** |

Table 2: Quantitative results of MoCA on the StyleGAN2 base architecture. Following the setting in StyleGAN2-ADA's original paper (Karras et al., 2020b,a), we use adaptive discriminator augmentation (ADA) to train the discriminator. We use 64x64 image size when performing experiments on StyleGAN2 and MoCA-StyleGAN2 on ImageNet-100* and COCO-300* and 256x256 image size synthesis on Animal Face Dog and Obama dataset. Consistent improvements when adding MoCA layer are observed despite the small image size and datasets.

| Generator Architecture | CIFAR-10 | | Generator Architecture | Animal Face Dog | | Obama | | ImageNet-100 | | COCO-300 | |
|---|---|---|---|---|---|---|---|---|---|---|---|
| | FID ↓ | KID ↓ | | FID ↓ | KID ↓ | FID ↓ | KID ↓ | FID ↓ | KID ↓ | FID ↓ | KID ↓ |
| StyleGAN2 | 5.19 | 2.43 | FastGAN | 51.25 | 17.11 | 43.13 | 13.52 | 66.44 | 22.31 | 30.43 | 6.02 |
| SA-StyleGAN2 | 5.60 | 2.79 | SA-FastGAN | 51.17 | 16.57 | 38.93 | 9.37 | 56.69 | 16.93 | 29.66 | 6.31 |
| MoCA-StyleGAN2 (ours) | **4.68** | **1.39** | MoCA-FastGAN (ours) | **48.27** | **13.83** | **37.19** | **7.81** | **52.01** | **7.23** | **26.67** | **4.53** |

Table 3: Ablation study on the improvement of MoCA comparing with GAN employing standard self-Attention (Zhang et al., 2019a)

on the discriminator side do not affect MoCA's improvement, as FastGAN based models use DiffAug whereas StyleGAN2 based models employ the ADA technique.

## 4.2 ABLATION STUDIES

In this section, we present ablation studies for MoCA layer compared with self-attention alone and the importance of the momentum head design. Additional ablation studies on the design of memory clustering organization used in MoCA can be found in Appendix 8.1, and more visualization analysis can be found in Appendix 8.6.

**Self-Attention v.s. MoCA** On both StyleGAN2 and FastGAN, we compare the performance of MoCA with a standard self-attention module to verify that the performance improvement of our model is not only due to the self-attention block, as several previous works have highlighted the advantage of applying self-attention on GANs to capture non-local dependency (Zhang et al., 2019b; Brock et al., 2018). Table 3 shows that in spite of the baseline architecture, MoCA consistently outperforms the standard Self-Attention module, suggesting that MoCA module plays an irreplaceable role in our method.

**Importance of Momentum Update Mechanism** We also study the effect of different ways to update the concept encoder. Table 4 shows that the models' FID on multiple datasets generally improves when using momentum to update the concept encoder. With momentum, the FID of FastGAN on larger and diverse datasets like AnimalFace Dog and CUB is notably better. This supports our motivation for using a momentum-updated encoder to build a more generalized memory bank that contains broader information beyond the current training batch.

| Model | FID↓ | | |
|---|---|---|---|
| | AnimalFace Dog | Obama | CUB |
| MoCA with momentum | **48.27** | **37.19** | **25.66** |
| MoCA *w/o* momentum | 63.08 | 46.25 | 51.74 |

Table 4: Ablation study on FastGAN + MoCA with different update mechanism

## 4.3 PROTOTYPE CONCEPT ANALYSIS

Here, we demonstrate that MoCA can learn in distinct semantic concepts in an unsupervised manner, represented by the "grandmother cells" or their representatives, during adversary learning for image synthesis.

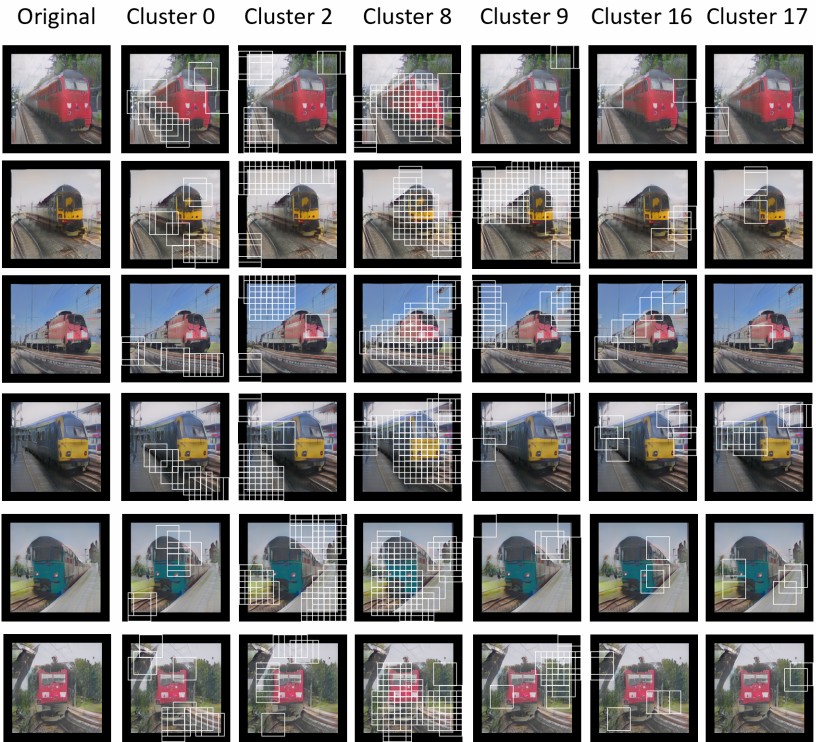

Figure 4: Visualizing Cluster Semantics in MoCA. We use MoCA-FastGAN model trained on MSCOCO-300 dataset for visualization (Lin et al., 2014). Each row is a generated image. We compute the cluster assignment for each hypercolumn in the layer where MoCA was installed and highlight the receptive fields of hypercolumns with white bounding boxes. We further group the visualization of the receptive fields by clusters (columns). For example, for column "Cluster 0", all white bounding boxes are the receptive fields of the hyper-columns that are modulated via Cluster 0's prototypes. We observe that different clusters have different semantics since their prototypes tends to modulate different semantic regions of the images (See discussion in Section 4.3).

**Semantic Concepts in MoCA**   We assigned a semantic cluster to each hypercolumn in the 16 x 16 feature map during a feedforward generation pass (cluster selection mechanism is described in Method Section 3.2). We label the receptive fields or projection fields of all the hypercolumns associated with each chosen prototype semantic cluster as white bounding boxes in the generated image, with some examples shown in Figure 4. We can observe that different clusters encode distinct semantic concepts. For examples, Cluster 0 tends to be associated with train rails. Cluster 2 covers area of uniform color such as sky and ground. Cluster 17 pays attention to the side of the train. Cluster 8 focus on the trains themselves, particularly the front. Along each row, we observe that the influence of MoCA for a specific image can be decomposed into different clusters. Some clusters are more popular (e.g. Cluster 0, 2, 8) while other clusters are used more selectively (for example, cluster 17 for the third image).

**Understanding Prototype Cells**   To understand the prototype cells that are cached inside each cluster of MoCA, we identified the generated image patch in the receptive field of any hypercolumn whose activation is closest to the prototype cell's memory in the assigned cluster of that hypercolumn from 100 image generations. Figure 5 show that the image patches closest to a prototype memory are similar visually, and that prototypes belonging to the same cluster are semantically related but distinct in visual features. We can see that different prototype cells are specialized as sub-parts of the semantic clusters they belong to. For example in Cluster 0, prototype 20 is a cached activation for rail or train tracks, but other prototypes include the top of the trains. Similar observations can also be made for Cluster 2 and 8 (Figure 5). These three clusters are queried frequently and therefore likely contain diverse but semantically related prototype memories. The prototype memories resemble the "visual concepts" of (Wang et al., 2017) and can potentially be used as visual concepts to implement hierarchical compositional system for image synthesis.

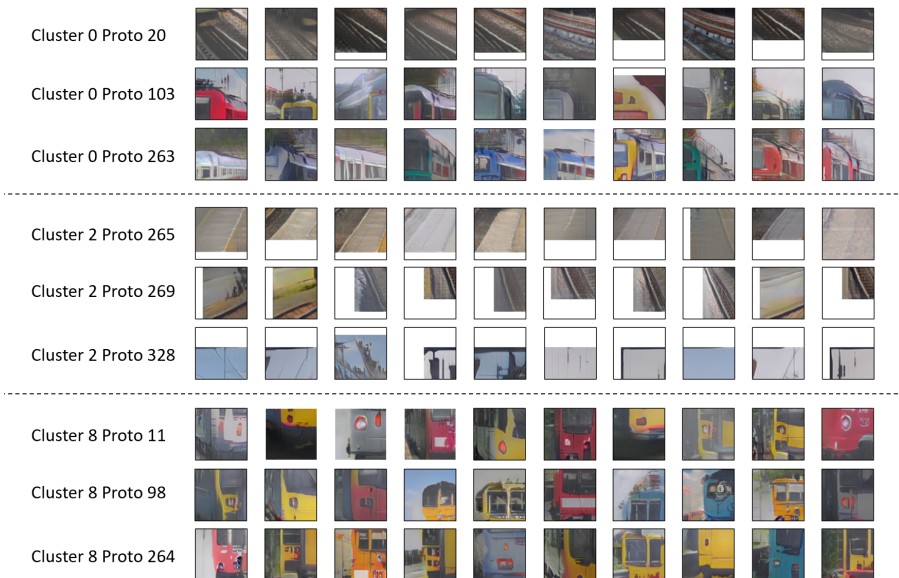

Figure 5: Visualizing different prototypes. Each image patch shown above is cropped based on the receptive field of a hyper-column. For each prototype inside MoCA's memory (each row above), we find hyper-columns whose activation will be largely modified from that prototype ("largely" if they are similar) during attention process and crop their corresponding receptive field in the generated images based on the convolution architecture (Details refer to the Appendix).

## 4.4 Robustness against noise

We evaluate the noise robustness of our proposed memory concept attention module and compare it against the standard self-attention module. The evaluation is conducted on FastGAN models trained without any injection of noise into the intermediate layers. In the evaluation stage, we inject Gaussian noise of different magnitudes with variance from 0 to 1 into the input feature map of the MoCA layer and standard self-attention layer. Our results (Table 5) show that, without the help of external memory, self-attention is more vulnerable to the noise attack, while MoCA is generally less sensitive to noise. We suspect that with MoCA, the feature map can attend to previously-stored concepts in the memory bank to retrieve noise-free part information, which alleviates the influence of noise perturbation. We also find that, no-cluster MoCA (All prototypes belong to the same cluster and jointly inference every activation $A_{ij}$, i.e. M=1) is more robust under higher levels of noise. We hypothesize that the feature map can attend to more concepts in no-cluster MoCA, increasing the chances of obtaining a correct bias.

| Model | Noise level | | | | |
|---|---|---|---|---|---|
| | 1 | 0.7 | 0.5 | 0.3 | no noise |
| Self-attention | 163.71 (±7.68) | 114.17 (±5.10) | 75.48 (±4.32) | 45.42 (±3.61) | 26.65 (±0.33) |
| MoCA (*w/o cluster*) | **79.71**(±2.79) | **63.84** (±1.68) | **54.63** (±1.02) | 45.82 (±1.28) | 37.04 (±0.31) |
| MoCA (*with cluster*) | 117.38 (±4.32) | 83.86 (±4.40) | 59.24 (±2.63) | **38.93** (±3.71) | **23.98** (±0.12) |

Table 5: Image synthesis FID by MoCA-FastGAN on CUB under different levels of noise injection.

## 5 Conclusion

In this paper, we introduced a module called MoCA, which can be inserted into any layer of a hierarchical neural network such as a GAN to improve image synthesis. MoCA caches part prototype memories over time, dynamically updated using a momentum encoder, and has the ability to modulate the continuous response of intermediate layers via an attention mechanism. In contrast to earlier memory bank studies which stored representation of the entire images as prior in the latent space for object recognition, the prototypes in MoCA are remembered parts and sub-parts of objects, that can be extracted at every level of the hierarchy, and flexibly composed for image synthesis. Our work was inspired by the discovery of a diverse set of highly selective complex feature detectors in V1, which correspond to the prototype cells in MoCA. The selection of different prototype clusters via semantic cells could correspond to switching circuits mediated by inhibitory neurons. While we do not claim that there is any serious correspondence between our model and neural mechanisms or circuits, the effectiveness of MoCA might still provide some insights to the roles of these "grandmother neurons" at a functional level.

## 6    ETHICS STATEMENT

This study investigates a novel proposal for understanding the computational advantages of having prototype-like "grandmother" neurons in the visual cortex, namely, that these prototype neurons can serve as auxiliary memory priors to modify both the embedding space of the self-attention mechanism as well as the activation patterns of neurons during the image synthesis process. While the motivation is for advancing our understanding of contextual modulation in neural systems, our work also results in a novel mechanism for contextual modulation in deep learning systems, particularly in image generation, and thus has a broader positive impact on machine learning, particularly on few shot learning. However, image generation technology can be misused for misinformation, with adverse social impact, e.g. the creation of "deepfakes." On the other hand, advancing our understanding of image generation is also crucial for combating deepfakes and other misuses of image generation technology by bad actors.

## 7    REPRODUCIBILITY STATEMENT

Our project code can be found at https://github.com/Crazy-Jack/MoCA_release.

## ACKNOWLEDGEMENT

This work was supported by an NSF grant CISE RI 1816568 awarded to T.S.L. This work is also partially supported by the start-up fund provided by CMU Mechanical Engineering Department.

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

# 8 APPENDIX

## 8.1 ABLATION: IMPORTANCE OF MEMORY CLUSTERING ORGANIZATION

| # Cluster | Size of Cluster | FID ↓ | KID ↓ |
|-----------|-----------------|-------|-------|
| 1 | 512 | 5.30 | 2.53 |
| 1 | 8192 | 4.76 | 1.52 |
| 3 | 256 | 5.10 | 1.96 |
| 20 | 256 | 4.91 | 1.61 |
| 20 | 1024 | 4.90 | 1.56 |
| 32 | 256 | **4.68** | **1.39** |

Table 6: StyleGAN2 on CIFAR-10 with different architectures. The first column denotes the number of cluster in the memory, 1 indicates no clustering mechanism is applied. The second column denotes the number of concept vectors attributed to a single cluster.

The second aspect we study is the impact of the clustering mechanism on the overall model performance. We find that for MoCA, a single concept pool with sufficiently large size, can still has similar performance compared against MoCA with multiple clusters. As shown in Table 6, the performance of no-cluster MoCA containing 8192 concepts inside, is at the similar level as MoCA with 32 clusters. We hypothesize this is because, although there is no clustering mechanism to enforce the memory bank learning diverse concepts in no-cluster MoCA, given a very large pool of concepts, it is still able to memorize diverse and useful concepts. Despite the finding that no-cluster MoCA can have similar performance to cluster MoCA with a sufficiently large pool, we argue that using the clustering mechanism is still superior. Uusing the clustering mechanism greatly reduces the amount of vectors involved in the attention calculation, which enables us to build a larger memory bank for complex datasets.

## 8.2 MOCA ON LS-GAN

To strengthen our conclusion on MoCA, we have also conducted experiments on the least-square GAN (LS-GAN)(Mao et al., 2017) framework which has a different loss and architecture compared to FastGAN and StyleGAN. We found MoCA can still provide a certain amount of improvement, consolidating the advantages of the MoCA layer.

| Generator Architecture | Discriminator Augmentation | Animal Face Dog | | Obama | | Cifar-10 | |
|------------------------|----------------------------|-------|-------|-------|-------|-------|-------|
| | | FID ↓ | KID ↓ | FID ↓ | KID ↓ | FID ↓ | KID ↓ |
| LS-GAN | DiffAug | 135.83 | 109.32 | 180.35 | 226.45 | 56.37 | 4.21 |
| MoCA-LS-GAN (ours) | DiffAug | **122.89** | **91.14** | **172.97** | **206.38** | **50.48** | **3.86** |

Table 7: Quantitative results of MoCA on LS-GAN based architecture. All experiements in this table are conducted using 64x64 size images. Results suggest a consistent improvement of MoCA layer.

## 8.3 LIMITATION AND FAILURE

Despite MoCA can bring in consistent improvement on most of the dataset with limited data, we also observe case where MoCA is less effective. We found that adding MoCA on StyleGAN for training Grumpy-cat leads to a slight performance decrease and leads to small improvement on FastGAN. This suggests that when the underlying dataset is less diverse and the base network is quite large, MoCA cached concepts can potentially cause more distraction during generation, leading to performance setbacks.

| Generator Architecture | Discriminator Augmentation | Grumpy Cat | |
|------------------------|----------------------------|-------|-------|
| | | FID ↓ | KID ↓ |
| FastGAN | DiffAug | 26.65 | 5.71 |
| MoCA-FastGAN (ours) | DiffAug | **25.68** | **5.18** |
| StyleGAN2 | ADA | **24.17** | **3.82** |
| MoCA-StyleGAN2 (ours) | ADA | 24.65 | 3.98 |
| LS-GAN | DiffAug | 99.92 | 88.43 |
| MoCA-LS-GAN (ours) | DiffAug | **83.82** | **73.22** |

Table 8: Quantitative results of MoCA on Grumpy-cat dataset.

## 8.4 ALGORITHM

Here, we present the algorithm that is presented in Figure 1 and Figure 2 in the main text.

---

**Algorithm 1:** MoCA + Self-Attention in the Generator

---

**Result:** Updated activation $\hat{A}$ after MoCA and Self-Attention; Updated Memory $P$.
$A \in R^{n \times c \times h \times w} \leftarrow$ from previous layer;
$\{\theta(A), \phi(A), \psi(A)\} \in R^{n \times \tilde{c} \times h \times w} \leftarrow$ transform into $\tilde{c}$ dimensional space via 1x1 conv;
# Memory Concept Attention;
**for** *every spatial column $\boldsymbol{a} \in R^{\tilde{c}}$ in $\theta(A)$* **do**

    Choose prototype semantic cell $K_i$ where $i \leftarrow \arg\min_i \|K_i - \mathbf{a}\|_2^2$;

    $\mathbf{E}^{(i)} \in R^{\tilde{c} \times h} \leftarrow$ Retrieve Prototype Component Cells for $K_i$ from memory;

    $\boldsymbol{s} \leftarrow [\mathbf{E}^{(i)}]'\mathbf{a}$          # compute dot product with memory;

    $\boldsymbol{\beta} \leftarrow \frac{exp\{\boldsymbol{s}\}}{\sum_{l=1}^{T} exp\{\boldsymbol{s_l}\}}$;

    $\boldsymbol{h}_m \leftarrow \mathbf{E}^{(i)} \boldsymbol{\beta}$          # $\boldsymbol{h}_m \in R^{\tilde{c}}$;

**end**
$\mathbf{H}_m \leftarrow$ Combining $\boldsymbol{h}_m$          # $\mathbf{H}_m \in R^{n \times \tilde{c} \times h \times w}$;
# Self-Attention;
$\boldsymbol{S} \leftarrow [\theta(A)]^T \phi(A)$;
$\hat{\boldsymbol{S}} \leftarrow softmax(\boldsymbol{S})$;
$\mathbf{H}_s \leftarrow \psi(A)\hat{\boldsymbol{S}}$          # $\mathbf{H}_s \in R^{n \times \tilde{c} \times h \times w}$;
$\boldsymbol{H} \leftarrow \mathbf{H}_m \oplus \mathbf{H}_s$;
$\hat{A} = \gamma O(\boldsymbol{H}) + A$;
# Memory Update;
$A_m \leftarrow \tilde{\phi}(A)$;
**for** *every column $\boldsymbol{a}_m \in R^{\tilde{c}}$ in $A_m$* **do**

    Choose prototype semantic cell $K_i$ where $i \leftarrow \arg\min_i \|K_i - \mathbf{a}_m\|_2^2$;

    Randomly update one column in $\mathbf{E}^{(i)}$ with $\mathbf{a}_m$;

**end**
$\tilde{\phi}_\theta \leftarrow \tilde{\phi}_\theta * (1 - m) + \phi_\theta * m$          # momentumly update the $\tilde{\phi}$

---

## 8.5 EXPERIMENTAL DETAILS

In this section we provide details for how we implement MoCA on FastGAN and StyleGAN 2 respectively. For reference, we provide our code and image ID list of MSCOCO, ImageNet subset used in the experiments at https://github.com/Crazy-Jack/MoCA_release. All the datasets used in experiments are publicly available and for non-commercial usage only. Among them, CIFAR-10[†] is attributed to Krizhevsky (2009), Obama, Animal Face, Grumpy Cat[‡] are attributed to Zhao et al. (2020), and Caltech-UCSD Birds-200-2011[§] (CUB) is attributed to Wah et al. (2011).

We insert MoCA layer between the last but one residual block and the last residual block of the generator. Given the input feature map $\mathbf{A}$ of the last but one residual block $f^{(l-1)}$, the last residual block $f^{(l)}$ and MoCA module $\Phi$, suppose the original image generation procedure on the last two blocks is:

$$\hat{\mathbf{x}} = f^{(l)}\left(f^{(l-1)}(\mathbf{A})\right). \tag{3}$$

Then the feedforward procedure including MoCA can be defined as:

$$\hat{\mathbf{x}} = f^{(l)}\left(\Phi\left(f^{(l-1)}(\mathbf{A})\right)\right), \tag{4}$$

---

[†] https://www.cs.toronto.edu/~kriz/cifar.html
[‡] https://hanlab.mit.edu/projects/data-efficient-gans/datasets/
[§] http://www.vision.caltech.edu/visipedia/CUB-200-2011.html

where $\hat{x}$ is the generated image. Please note that, to enable that MoCA can be flexibly integrate into arbitrary layer, MoCA will not change the channel numbers and resolution of the feature map.

### 8.5.1 FASTGAN

**Network Architecture and Optimization**  We use exactly the Fast-GAN architecture Liu et al. (2021) as the backbone. MoCA layer is inserted after the 64x64 resolution layer for all experiments run for CUB, AnimalFace Dog, Obama and Grumpy-cat. We use Adam optimizer for training and 0.0002 for learning rate with $\beta_1 = 0.5$ and $\beta_2 = 0.999$. Differentialbe augmentation is used during training. We utilize the loss proposed in (Liu et al., 2021) for stable and optimal training. For all our experiments, 100K iteration is run to perform fair comparsion.

**Computational Resource**  With FastGAN and MoCA, it takes about 15 hours to train 100K iteration on a single GTX-1080Ti GPU.

### 8.5.2 STYLEGAN 2

**Network Architecture and Optimization**  We use exactly same architecture from Karras et al. (2020a) as the backbone. We use Adam (Kingma & Ba, 2017) optimizer with a learning rate of 0.0025 and $\beta_1 = 0, \beta_2 = 0.99$. We adopt the same training configuration as the Official PyTorch implementation of StyleGAN 2 (including Adaptive discriminator augmentation, path length regularization , lazy regularization (Karras et al., 2020b), and non-saturating logistic loss (Goodfellow et al., 2014)). For the training on the large dataset (CIFAR-10), we train models for 200k iterations of gradient updates, and 100k iterations for few-shot datasets.

**Computational Resource**  We train StyleGAN2-based model with 4 RTX-2080Ti GPUs each run. The training of each model takes roughly 20 - 35 hours to complete, depending on the image resolution.

### 8.6 MEMORY CONCEPT ANALYSIS

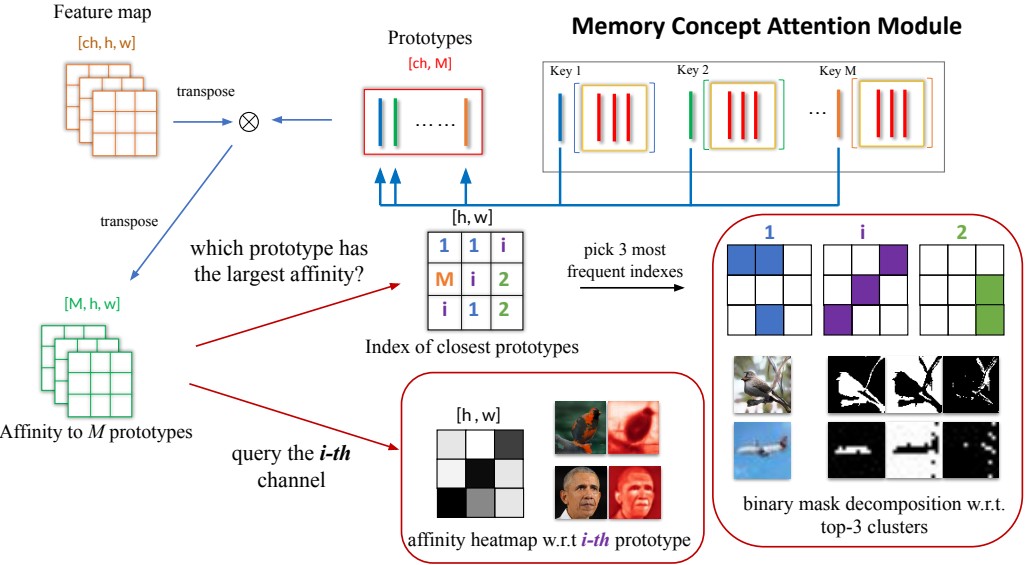

Figure 6: Schematic of affinity map visualization procedure

We generate 50,000 images from random noise for CIFAR-10, and 1,000 images for 100-shot Obama. For each feature map and specific concept cluster $i$, we count how many pixels' closest cluster (measured by the magnitude of the dot product) is $i$, and repeat this process for all clusters. Then we sum the count and pick top-K clusters which have the most pixels, and visualize their affinity

heatmap. To interpret the vector embeddings (concepts) stored in each concept cluster, we perform three kinds of visualization: affinity heatmap w.r.t. different concept clusters, images' binary mask w.r.t. different concept clusters, and t-SNE visualization (van der Maaten & Hinton, 2008) of the feature map before and after attending with memory concept. The detailed information retrieval process is shown in Figure 6. We use StyleGAN2+MoCA's results for visualization, as they perform the best on most of the dataset.

**Semantic concepts** In Figure 7a and Figure 7b, we show the resulting affinity maps computed as discussed above (detailed schematic of visualization procedure is shown in Figure 6). Note that we select clusters and images for demonstration purpose. Results on CIFAR10 using MoCA-StyleGAN (32x32 resolution) and Obama dataset using MoCA-FastGAN (256x256 resolution) are acquired. We find that these affinity maps demonstrate that concepts formed during MoCA training cover crucial and diverse patterns for image synthesis. As shown in Figure 7a and Figure 7b, patterns of different clusters are usually semantic meaningful and very distinct. On CIFAR-10, we notice that cluster 11 stores concepts related to the sky, while cluster 13 stores concepts related to the frame of trucks. Cluster 7 favors the white space while cluster 18 has a "crush" on animals' body and head. Similarly, on the other dataset, different clusters also focus on very different patterns. For instance, on 100-shot Obama, cluster 3 stores the concepts closely related to ties, and cluster 9 contains concepts related to the light reflection on the face.

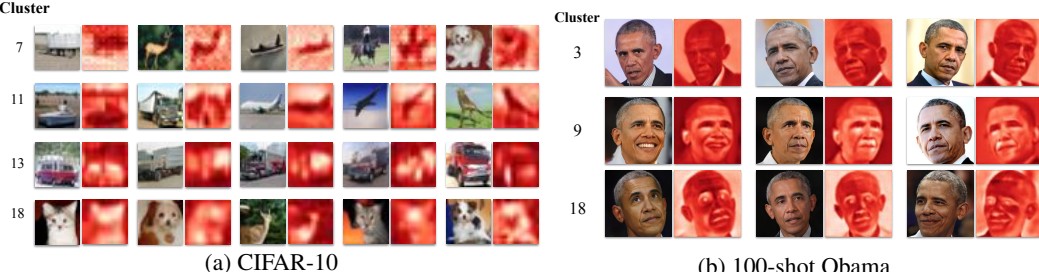

(a) CIFAR-10          (b) 100-shot Obama

Figure 7: Affinity heat-map w.r.t. different concept clusters. White: strongest affinity; Dark red: zero affinity. **Left (a):** MoCA-StyleGAN2 on CIFAR10. **Right (b):** MoCA-FastGAN on 100-shot Obama dataset. We can observe that Different cluster associated with different semantic meanings although not trained with category supervision.

**Prototype attention** We study the internal dynamics of MoCA by using t-SNE to visualize the changing trend of feature map before and after attending with concepts in the memory. We generate 10,000 images from random noise on CIFAR-10 and extract the feature maps before and after MoCA layer. Figure 9a shows that the activation patterns (blue dots) from previous layer, after the $\theta$ transform, are effectively influenced by MoCA (green box in Figure 1) into a representation (red dots) that are pulled towards or stretched by certain familiar prototype exemplars (green dots).

**Image synthesis as concepts assembling** By visualizing the affinity to the prototype of each pixel in the feature map, we acquire a high-level understanding of what concepts different clusters are trying to capture. We can also further investigate the relationship between different concepts and the image synthesis process. We decompose the image into different binary masks with regard to top-3 clusters. The decomposition is shown in Figure 8. Concretely, given a specific cluster $C_k$ and a feature map $F$, we generate the binary mask w.r.t. $C_k$ by querying whether pixels'($F_{ij}$) closest cluster (highest affinity) is $C_k$ or not, where white denotes true and black denotes false. We pick the top-3 clusters for each image and visualize their binary mask as shown in Figure 8. For most images, concepts from the top-2 clusters are usually related to foreground and background information , while the third cluster contains concepts related to high-frequency details in an image. This demonstrate that, with MoCA, image synthesis can also be viewed as a process of retrieving different concepts from memory and assembling them together to create a realistic image.

**Implicit influence of MoCA the attention mechanism** Interestingly, we also observe that having the MoCA module can in turn sharpen the functional activities of the horizontal interactions. To visualize the functional connectivity of the horizontal connections, we plot the normalized rank-order

affinity score of the self-attention map in the 16x16 resolution layer of the converged StyleGAN2 trained on CIFAR10.

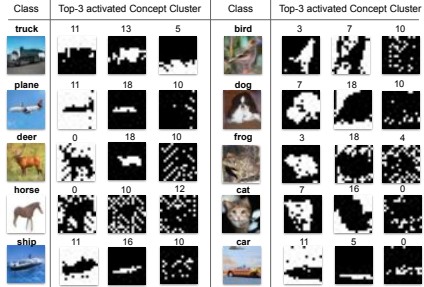

Figure 8: CIFAR-10 image binary decomposition w.r.t. their top-3 activated clusters.

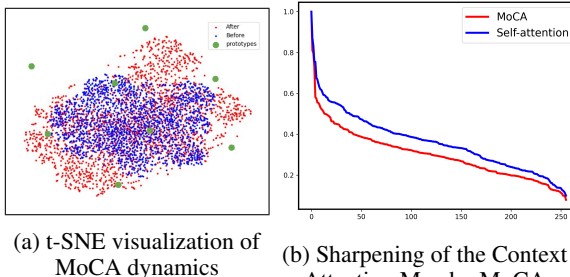

(a) t-SNE visualization of MoCA dynamics

(b) Sharpening of the Context Attention Map by MoCA

Figure 9: Modification of the activation representation and Context Attention Map by MoCA.

In Figure 9b, the purple curve represents the rank-ordered curve without installing the MoCA and the red represents that with MoCA. Clearly, MoCA sharpens the functional connectivity among the horizontal connections, even though we do not explicitly modify the self-attention process in the MoCA layer, indicating that our prototype memory is implicitly influencing the horizontal connections.

## 8.7 METRIC EVALUATION

For all models, we evaluate metric using the last 3 snapshots of models (with an interval of 10k gradient updates) and report the best one. We train all StyleGAN-based models for 200k iterations and all FastGAN-based models for 100k iterations. To measure the model performance on Animal-Face Dog (Si & Zhu, 2012), 100-Shot-Obama (Zhao et al., 2020), ImageNet-100 (Russakovsky et al., 2015), COCO-300 (Lin et al., 2014), we sample 3,000 noise vectors from standard Gaussian and calculate FID and KID score of the generated images. On CIFAR10 (Krizhevsky et al., 2009) and Caltech-UCSD (Welinder et al., 2010), we sample 50,000 respectively to adjust for number of training data.

## 8.8 GENERATED IMAGES AND THEIR NEAREST NEIGHBOR IN THE DATASET

Although MoCA proposes a memory module, our improved results are not coming from trivially memorizing the training images. First, the metrics we are using (FID and KID scores) take the generation diversity in to consideration. We sample over 3,000 or 50,000 images during the evaluation process, however the entire training dataset only contains hundreds. Second, to qualitatively test if MoCA only perform generation by memorizing the training images, we find the nearest neighbor of generated images based on the Learned Perceptual Image Patch Similarity metric (LPIPS) (Zhang et al., 2018). The feature extractor used here is a pretrained VGG16 (Simonyan & Zisserman, 2015). In Figure 10, we see that on the left is some images randomly chosen from the MoCA-FastGAN generation, and on the right is top-3 closest training images based on the LPIPS (Zhang et al., 2018) similarity metrics. We can clearly observe that the generated images are different from the training images. And more interestingly, we can see that the generated images indeed incorporate some parts level information and combine those parts level examples to compose new images. For example, the generated train example on the second row (left) may have the similar train head as the second training image, but resembles a similar train side view (the shape) as the first training image. The overall direction of the rails also have looks similar as the first image but it replace the rain shelter on the boarding platform with an open sky with some electric cables hanging over. This further consolidates with the observation made in Section 4.3 that MoCA works at modulating the parts level generations. Overall, the results in Figure 10 suggests MoCA's generations are not only memorization of the training images but rather combine different parts to compose a new instance.

**Generated image** | **Top 3 Nearest Neighbors in the dataset**

Figure 10: Some randomly generated images and their corresponding nearest neighbor in the dataset. Left: Generated images from MoCA-FastGAN models. Right: Its top-3 nearest neighbors in the training dataset with the similarity score rank from left to right as high to low. The similarity is measured by perceptual distance (Zhang et al., 2018)

## 8.9 VISUALIZATION DETAILS

**Details of Receptive Field Localization** In order to generate the receptive field of each hyper-column (white bounding box in Figure 4), we calculate the receptive field according to the network architectures (Liu et al., 2021). The final image is of 256 x 256 resolution. For a single hyper-column in the layer we install MoCA (layer 16 x 16), there are 4 up-sampling layers (each with factor of 2) and 7 convolution layers (each has 3x3 kernel size, 1 stride and 1 padding), resulting in a 66 x 66 receptive field size. The location is traced based on the recursive derivation of the convolution and up-sampling operations (Goodfellow et al., 2016).

**Details of Visualizing the Prototypes** Since our prototypes only influence the activation via attention, it is not straightforward to visualize a prototype. However, since our prototypes are used to modulate the hyper-column weighted by the normalized similarity between the $\theta(A_{ij})$ and the prototype vector, we can consider hyper-columns that have high **cosine similarity** with the prototype vector in the projected space as an approximation of that prototype because if they are similar, there will be a large weight during attention process. Since each hyper-column would correspond to a patch generation in the image, we can visualize the corresponding patch and get a sense of what the individual prototype cell is representing for.

