# OpenReview forum: "Prototype memory and attention mechanisms for few shot image generation"
_ICLR.cc/2022/Conference — ICLR 2022 Poster_

### Official Review · Reviewer_9zZN · 2021-10-31

**Correctness:** 4
**Technical Novelty And Significance:** 3
**Empirical Novelty And Significance:** 3
**Recommendation:** 8
**Confidence:** 3

**Details Of Ethics Concerns:**

No concern.

**Main Review:**

Strengths
- Novelty. A novel prototype-based layer which significantly improves the respective augmented image generation models: FastGAN, StyleGAN2.
- Significance. The few-shot image generation is an important research topic.
- Relevance. Will be of interests to ICLR community, especially the vision researchers.
- Experimental Design. The performance improvements of adding MoCA layer to  two state-of-the-art architectures, FastGAN and StyleGAN2 were validated with 6 datasets of few-shot image synthesis: Animal-Face Dog, 100-Shot-Obama, ImageNet-100, COCO-300, CIFAR10 and Caltech-UCSD Birds (CUB).
- Experimental Results. The consistent and somewhat significant improvements over the two baselines are impressive.
- Suitable ABLATION STUDIES were done.
- Analysis using visualization were performed to better understand the model.
- Writing is clear and well-structured.

Weaknesses
- Minor. Only 2 baselines were used to validate the novel design. The paper will be significantly strengthen with experiments on more baseline models.
- Minor. The modeling of the proposed method: MOCA has very limited correspondence in neuroscience. This is despite the paper's claim of being "inspired by neuroscience discoveries". As paper did not claim to be based on neuroscience, this is not a major concern.

**Summary Of The Paper:**

Paper proposes a novel prototype-based memory modulation layer (MoCA) to improve the generator network of a GAN. The target problem is few-shot image generation. Memory is arranged hierarchically into prototype semantic cells and prototype component cells. This design is loosely inspired by the recent discovery of "grandmother cells" in V1.

Experimental results show that in terms of FID score, using FastGAN base architecture, MoCA can bring 5.8% improvement on Animal Face Dog, 13.8% improvement on Obama, 21.7% improvement on ImageNet-100 and 12.4% improvement on COCO-300 dataset when using FastGAN as the baseline models. With StyleGAN2 base architecture, there was an 5.1% improvement on Animal Face Dog dataset, 8.1% improvement on Obama dataset, 14.1% improvement on ImageNet-100 dataset and 17.3% improvement on COCO-300 dataset.

**Summary Of The Review:**

A well-written paper with novel and significant contributions to an important research topic. The design is clearly explained and the experimental results are excellent. While there are a couple of minor issues, the overall quality is sufficiently high for an acceptance by ICLR.

---

> ### Author Response · Authors · 2021-11-22
> **Response to Reviwer 9zZN**
>
> We thank the reviewer for the very positive comments. We hope that the updates we added to the paper have further strengthened it. Below we would like to address the concerns.
>
>
> **(Concerns about baseline numbers)**
> We appreciate the reviewer's concern. FastGAN and StyleGANv2 are chosen because they are the strongest image generations models in either unconditional few-shot image generation (FastGAN) or the image generation domain in general (StyleGANv2) in terms of architecture design. We do include another baseline model (LSGAN) to further test the generalization of MoCA. Although the LSGAN backbone is weaker compared to FastGAN and StyleGANv2, we still observe that MoCA can improve upon the baseline model by a certain amount (See Appendix 8.2 for details).
>
> **(Concerns about neuroscience correspondence)**
> We thank the reviewer for sharing this concern. Our work was inspired by the neuroscience findings but the correspondence at best is at a functional level. We don't intend to claim this is a neural model that has strong correspondence to biological evidence. Our work is rather a computer vision method that was designed with neuroscience questions in mind. Although the exact biological correspondence may still be not obvious at the moment, the performance improvement in the image generation task could spark ideas about the potential computation roles of the sparse complex feature detectors in the hierarchical compositional visual systems. We have updated our manuscript to make the connection and limitations more clear.

---

### Official Review · Reviewer_4n7p · 2021-11-02

**Correctness:** 3
**Technical Novelty And Significance:** 2
**Empirical Novelty And Significance:** 3
**Recommendation:** 5
**Confidence:** 4

**Main Review:**

Strengths:
The idea of using prototye memory concept is interesting, although it is not a new idea. For example, the prototypical network for few shot learning [1]. The authors showed that the proposed MoCA module can be used to enhance the geterator in several GAN based few-shot image generation models.

Weakness:
The authors should make the submission more clearer. The main concerns are listed below:
1) It is still an open question that how the "grandmother cell"  (if exists) codes information in the brain, e.g., population coding, sparse coding, or in other forms. The sparse activity of V1 neurons in the monkey brain doesn't mean they are grandmother cells. Activities of neurons in deeper layers (L3-5) may not be sparse. The authors should be more careful about this.
2) the memory bank idea has been proposed in many other works (the implementation detailed may be different), the authors should compared the difference of their work and others.
3) What do the prototype semantic cells and the prototype component cells correspond to in the neocortex? Is there a correspondence, or it is just for network design? Why not only use  $K_i$ to represent the prototype memory?
4) When transform A to a low-dimensional space, why do you use three functions $\theta(\cdot)$, $\phi(\cdot)$ and $\psi(\cdot)$? I only find $\phi(\cdot)$ in Fig.2. Also, in the main text, sometimes the authors use $\Phi$ and $\Psi$. All the terms should be defined clearly at the begining.

[1] Snell, J., Swersky, K., & Zemel, R. S. (2017). Prototypical networks for few-shot learning. arXiv preprint arXiv:1703.05175.

**Summary Of The Paper:**

The authors proposed a Memory Concept Attention mechanism to improve few-shot image generation quality. They showed that the prototype memory bank improves image synthesis quality, learns interpretable visual concept clusters and improves the robustness.

**Summary Of The Review:**

The authors carried out adequate experiments to show the proposed module (MoCA) is can be used to improve few-shot image generation quality and they linked the module to the grandmother cell in the brain. However, the memory bank idea is not new and the novelty is limited.

---

> ### Author Response · Authors · 2021-11-22
> **Response to Reviwer 4n7p - Part 2**
>
>
> **(Neuroscience correspondence, regarding the neural analog of semantic prototype cell, and Why not use Ki for prototype memory)**
>
> Our inquiry was inspired by the finding of the super-sparse code of neurons in the superficial layer of V1. We wish to know the computational advantages of having neurons with such high response sparsity and selectivity in the feedforward (analysis) / feedback (synthesis) paths in the hierarchical visual system. We used a computer vision task to explore this question at a functional level. Note that we don't claim our work is a neural model that resembles deeply the biological details, but rather an exploration of potential computational roles of these recently found "grandmotherly" neurons in a hierachracial visual system. Overall our model is a computer vision model that is designed with some neuroscience properties in mind. The benefit MoCA brings on the computer vision task can potentially hint at our questions about the computational advantages of such neurons. We clarified the connections and the limitations in the revised manuscript (abstract, introduction, and conclusion).
>
> As to the questions, "What do the prototype semantic cells and the prototype component cells correspond to in the neocortex? Is there a correspondence, or is it just for network design? " In our mind, the prototype component cells correspond to the super-sparse cells found in a superficial layer of V1 that motivated our study. The prototype semantic cells, if you allow us to speculate, might be a type of inhibitory neurons (SOM) that perform circuit switching to select a certain subset of prototype cells to participate in contextual modulation.
>
> The prototype semantic cell represents more abstract information and does not have all the fine detailed information to provide the appropriate contextual modulation. The prototype semantic cells are responsible for selecting an appropriate set of prototype cells to participate in the attention process. The group of prototype (grandmother) cells needs to work together to provide the appropriate contextual modulation.
>
> **(Questions about $\theta$, $\phi$ and $\psi$)**
> We appreciate the reviewer's question. We clarify in the updated main architecture figure (Figure 1) that our proposed method contains 2 paths, one is Memory Concept Attention (MoCA) and the other is Self-Attention (SA). MoCA and SA share the same projection head. The use of 3 heads is followed as the convention in the Self-Attention/non-local network literature: $\theta$ is considered as "query", $\phi$ are considered as "key" module in SA [3]. $\tilde{\phi(\cdot)}$ is $\phi$'s corresponding momentum update version, through which we cache $\tilde{\phi(A_{ij})}$ into the memory. $\psi$ corresponds to the "value" in SA literature [3]. We update Figure 1 and Figure 2 for a more clear demonstration of the network architectures. We do not intend to use capitalized $\Theta$, $\Phi$, and $\Psi$ and correct them in the updated manuscript.
>
>
>
>
>
> [1] He, Kaiming, et al. "Momentum contrast for unsupervised visual representation learning." Proceedings of the IEEE/CVF Conference on Computer Vision and Pattern Recognition. 2020.
>
> [2] Snell, J., Swersky, K., & Zemel, R. S. (2017). Prototypical networks for few-shot learning. arXiv preprint arXiv:1703.05175.
>
> [3] Zhang et al. Self-Attention Generative Adversarial Networks. ICML 2019.

---

> ### Author Response · Authors · 2021-11-22
> **Response to Reviwer 4n7p - Part 1**
>
> We thank the reviewer for the constructive feedback. Below we would like to address the concerns.
>
>
> **(Concerns about grandmother cells in neuroscience)**
> We thank the reviewer for the advice. We made it clear in the abstract and the introduction of the updated manuscript as follows, "We called these highly selective sparse-responding feature detectors 'grandmother neurons' to highlight their possible explicit encoding of prototypes, even though in reality, a prototype is likely represented by a sparse cluster of neurons in the brain. Neurons in different layers of each visual area exhibit different degrees of response sparseness, complementing one another in various functions. "
>
>
> It is worth noting that even in our model, there is also a variety of neural codes, ranging from distributed, sparse, and "grandmotherly." The code of the neurons in the feature layers in the GAN for example are quite distributed. Distributed codes provide greater flexibility and power in analysis and synthesis. Further, while a "grandmother cell" in MoCA might be coding an explicit prototype memory or visual concept, in MoCA, multiple grandmother cells, not just one, in the same semantic cluster would be activated by any particular input, and their codes are combined (weighted by softmax) to generate the needed contextual modulation to the activation patterns for downstream processing.
>
>
>
>
>
> **(Comparison to existed ideas on memory bank and prototype learning)**
> We appreciate the reviewer's suggestion. We add additional comparison discussion for the memory bank ideas in the literature under the related work section titled "Prototype Memory Mechanism". Indeed memory bank is not a completely new idea. However, most of the work is using memory bank at instance level [1] but our method proposes to use the memory bank at intermediate representation level (parts level). By having a memory bank at the parts level, our system incorporates the idea of a compositional system and reconfigurable parts naturally. We show that this is particularly valuable during the few-shot image generation task.
>
> In addition, we also discuss our difference with the existed prototype ideas like [2] under the related work section titled "Few Shot Prototypes Learning" in the updated manuscript. The main difference lies in the prototype forming level and how we utilize the prototypes. The prototypes in [2] are learned as representatives for different classes at the instance level, however, our prototype exists at the intermediate feature level (parts level). Also, our method utilizes an attention mechanism to softly bias the intermediate features using the cached prototypes during inference but [2] select the closest prototype in a discrete manner in order to perform category predictions at test time. Although we both involve prototype generation and usage during inference, our method with attention mechanism can smoothly modify the intermediate representation, hence can be applied to more challenging tasks like image generation in the few-shot setting. We refer further discussion to the revised manuscript Section 2.

---

> ### Author Response · Authors · 2021-11-30
> **Further Response to Reviewer 4n7p**
>
> Dear Reviewer 4n7p,
>
> Thank you again for your feedback. As the deadline for discussion is approaching, we would be happy to provide any additional clarifications that you may need.
>
> In our previous comments, we have carefully studied your comments and made updates to the revision as summarized below:
>
> - Revised the manuscript to clarify our connection to neuroscience and provide a discussion on the correspondence with grandmother cells, the role of semantic cells, and the rationale of choosing prototype memory.
> - Provided a further comparison with existing works utilized memory bank and prototype ideas.
> - Clarified the use of projection heads $\theta$, $\phi$, and $\psi$. Update the main architecture figures to make it more clear.
>
>
>
> Please let us know if you have any questions remaining. We would be happy to do anything that would be helpful in the time remaining!
>
> Thank you for your time!
>
> Best,
> Authors

---

### Official Review · Reviewer_JpF6 · 2021-11-02

**Correctness:** 3
**Technical Novelty And Significance:** 3
**Empirical Novelty And Significance:** 2
**Recommendation:** 5
**Confidence:** 3

**Main Review:**

This paper proposes a timely and interesting neural network memory mechanism, MoCA. The paper is fairly clear and well-written. However, the connections to neuroscience are weak (contra the introduction and discussion), and the image generation results raise some further questions that should probably be answered before recommending acceptance.

The authors’ central claim is that the proposed memory mechanism mechanism, which enables the network to memorize (transformed) activation vectors from images seen during training, similar to “grandmother cells” in human cortex, can enhance images generated by GANs. The connection to neuroscience here is tenuous at best — the authors draw inspiration from neuroscience, but I’m highly skeptical that the results are actually garnering any neuroscientific insight (to do that would probably require comparing these models to some form of brain recordings, but it is unclear how exactly that would be achieved with the current image-generation setup). So for the purpose of this review, I’m going to assume that this is simply a computer vision paper.

Quantitatively, the authors’ efforts seems successful: GANs trained using MoCA achieve better performance on few-shot learning than identical networks without MoCA. However, it is not clear that this comparison is entirely fair: do the StyleGAN2 and MoCA-StyleGAN2 (or FastGAN and MoCA-FastGAN) networks have the same number of parameters? This is not stated, but it seems like the MoCA layers will simply increase the parameter count, potentially inflating performance without revealing a real insight.

Relatedly, it seems important for the authors to test whether the MoCA layers are enabling the GANs to simply memorize the training images. In the example image figure (Figure 3), the authors should show the most similar image from the training set alongside each generated image.

The authors efforts to investigate the information learned in each memory cluster are important and interesting, but somewhat under-developed. In Figure 5 the authors show affinity maps for several memory clusters and images, and the results are somewhat convincing but also difficult to parse. I’m not sure how best to improve this analysis, but perhaps showing the same set of images for all the clusters would highlight the differences in cluster selectivity.

Unfortunately the second memory cluster analysis (Figure 6) is much more difficult to understand. Here it seems like the authors are interpreting the tiny generated patches in the left-hand-side of each image in an attempt to understand what the semantic selectivity of each cluster is. While it seems clear that these patches come from different parts of the image (e.g. the left-most cluster is definitely selecting for sky in each image), the generated patches are very tiny and not very distinctive. I was left unsure about the claims made based on this Figure. I would recommend that the authors re-work this to be more clear about what they are showing, but I am afraid that I can’t really offer any specific and substantive recommendations.


**Summary Of The Paper:**

In this paper, the authors propose a new “grandmother cell”-like memory mechanism for improving image generation performance in GANs. In short, this method clusters and stores activation vectors observed during training. Then at image generation time, activation vectors are augmented with the sum of the stored memories at the closest cluster. This seems to improve GAN performance for few-shot image generation tasks. The authors also visualize the learned memory clusters, which seems to reveal some semantic clustering.


**Summary Of The Review:**

This paper proposes an interesting memory mechanism, and shows that it seems to help in some circumstances. However, questions about the quantitative evaluation, qualitative evaluation, and interpretation analyses leave me unsure about how much improvement the method really offers.

---

> ### Author Response · Authors · 2021-11-22
> **Response to Reviwer JpF6 - Part 2**
>
>
> **(Memorizing examples from the training set)**
>
> Does MOCA simply remember example images for few-shot image generation? We have provided further justification and empirical analysis in the Appendix. MoCA is a module that can be installed in every layer in theory. In our current implementation, we put MoCA in an intermediate layer. Hence it only caches the parts level activations, analogous to visual concepts, but in a remapped space. The prototypes are clusters over the entire dataset, gathering part-level prototypes that can potentially be used as reconfigurable parts to synthesis novel images in a compositional framework. Such parts are useful for few-shot image generation because the training data is limited, and hence reconfiguration of the existed parts could ease the training process. However, in contrast to AndOr Image Grammar, our reconfiguration is softer, implemented using an attention mechanism.
>
>
> Empirically, we perform comparisons with the top 3 most similar training images (Based on pre-trained VGG features and cosine similarity metrics) in the Obama, MSCOCO-300, and Animal Face Dog dataset to demonstrate that our MoCA trained model is not memorization of the training set (See **Appendix Section 8.8** for demonstration and further discussion). We can qualitatively see that although they have some degree of similarity, they are **not** simply memorization of the train images.
>
>
> **(Concerns on MoCA cluster visualization)**
>
> To facilitate further understanding of the MoCA system, we update the manuscript to include two additional analyses (Figure 4, Figure 5, and Section 4.3). We answer the questions about (i) which semantic cluster is used during the attention process for different locations in the image? Is there any real semantics associated with it? (ii) What are the relationships between the prototype cells and their corresponding clusters? We provide detailed results and discussions in Section 4.3. Briefly, the results suggest that **(1)** Semantic clusters have semantic meanings, and different semantic clusters will bias different regions of the images based on the semantics. **(2)** Prototype cells are specialized sub-parts of the parent semantic concept clusters, suggesting MoCA may have the potential to facilitate a hierarchical compositional system in the image synthesis path.
>
> To make it concrete, consider a layer in which the prototype component cells are encoding face parts such as eyes, noses, and mouth. The prototype semantic cell will be coding "face-ness." The self-attention 1x1 function serves to map eyes, noses, and mouth to proximal locations in a low-dimensional transformed space, where these related concepts are clustered together and can be represented by their cluster center -- "face-ness" semantic cell. Note that this "face-ness" cell has the spatial scope of one hyper-column only and is different from the neurons in the preceding layer (upstream), which, with a larger receptive field or projective field, represent the entire face, with all the face parts in an appropriate spatial configuration. The prototype semantic cell is thus more like a circuit switch or grandmother selector. It represents an abstract semantic meaning like face-ness and is different from visual concepts or prototypes. The prototype semantic cell itself does not have all the fine detailed information to provide the appropriate modulation. We found that a subset of grandmothers turned on (or not turned off) by the prototype semantic cell need to work together to provide the appropriate contextual modulation.

---

> ### Author Response · Authors · 2021-11-22
> **Response to Reviwer JpF6 - Part 1**
>
> We thank the reviewer for the insightful comment. Below we would like to address the reviewer's concerns.
>
> **(Connection to neuroscience)**
>
> We thank the reviewer's cautionary note for the connection of our work to neuroscience. In fact, all three reviewers raised this issue, but all agreed that this is not a major concern, as we did not claim this is a neural model, just neural-inspired, and we agree that the paper should be judged as a computer vision paper. We have revised the paper to clarify this point.
>
> That said, we must say that our inquiry was genuinely triggered by the finding of the super-sparse code of neurons in the superficial layer of V1. The question we asked was, what could be the computational advantages on feedforward (analysis) / feedback (synthesis) paths in the hierarchical visual system for having this type of neurons with such high response sparsity and selectivity? We used a computer vision task to explore this question at a functional level. We did not claim that our model is an explicit neural model with a serious correspondence at an architectural or cellular level. However, we still believe our work is relevant to neuroscience at a functional level. From our perspective, CNN and GAN are relevant for understanding feedforward and feedback computation in the hierarchical visual cortex, respectively. The conceptual framework of analysis-by-synthesis and predictive coding are popular among cognitive and theoretical neuroscientists. The study of GAN and image generation are meaningful and potentially relevant for understanding the function of recurrent feedback. We tuned down our abstract and revised introduction to say, "Although our work is inspired by neurophysiological findings, we do not claim that the proposed MoCA module is a neural model. Rather, our goal is to explore the utility of prototype cells as memory priors in a standard computer vision image generation task, which might provide potential insights into the advantages of having these "grandmother" neurons in the visual cortex at a functional level."
>
>
> **(Parameter Inflation)**
> We thank the reviewer for bringing up this question. We address this concern with experiments from the results already documented in the original paper as well as new experiments we conducted in the rebuttal period.
>
> First, we do include a fair comparison with the baseline in our manuscript. Specifically, MoCA is an architecture layer that has two paths: one is conventional Self-Attention (SA), the other is a newly proposed prototype memory attentional path. Note that the projection head ($\theta(\cdot)$, $\phi(\cdot)$, $\psi(\cdot)$) of prototype memory path is shared with the Self-Attention, hence comparing to using self-attention alone, there's no **learnable** parameter inflation (learnable in the sense of gradient descent). Yet, Table 3 in the main text shows MoCA's improvement compared with Self-Attention.
>
> When it comes to comparison with original FastGAN and StyleGAN, our method only has an increase of a tiny percent of parameters in comparison to the overall **learnable** parameter numbers (about 0.1 % on FastGAN baseline and 0.6% on StyleGAN). Yet, the addition of a fixed parameter MoCA layer provides 5 % ~ 20 % performance improvement (measured on FID, see main text section 4.1).
>
>
> Furthermore, to completely control for the number of parameters, we added more parameters in terms of "features" in a layer to FastGAN (Adding 3 additional feature channels to the layer) so that they have the same number of parameters as MoCA + FastGAN. We found that the performance improvement it produced was minuscule (best FID improves less than 1% and overlaps with the normal size model for most of the time). Thus, the performance improvement observed in MOCA cannot be explained by parameter size only.

---

> ### Author Response · Authors · 2021-11-30
> **Further Response to Reviewer JpF6**
>
> Dear Reviewer JpF6,
>
> Thank you again for your feedback. As the deadline for discussion is approaching, we would be happy to provide any additional clarifications that you may need.
>
> In our previous comments, we have carefully studied your comments and made updates to the revision as summarized below:
>
> - Provided a discussion on how our work is related to neuroscience and revised the manuscript to clarify our work is not a neural model and MoCA only corresponds with neuroscience at a functional level.
> - Provided a discussion based on existing ablation and additional results to show that MoCA's performance improvements are not due to parameter inflation but the proposed mechanism itself.
> - Provided the top-3 nearest neighbors of the generated images to demonstrate that MoCA is not only remembering the train set but facilitating generalization by re-configuring parts-level information.
> - Add more interpretable visualization to demonstrate how MoCA works: Cache parts-level information which is then used to softly bias the corresponding activation.
>
>
> Please let us know if you have any questions remaining. We would be happy to do anything that would be helpful in the time remaining!
>
> Thank you for your time!
>
> Best,
> Authors

---

### Author Response · Authors · 2021-11-22
**General Response**

We would like to thank all reviewers for their feedback and constructive suggestions. We are very encouraged by reviewers' evaluation that our work is interesting ("timely and interesting" (JpF6), "concept is interesting" (4n7p), "novel and significant contributions" (9zZN)). We have taken into account reviewer feedback and provided an updated revision with a detailed response to each reviewer's suggestions and feedback. In the revised manuscript, we have highlighted our changes in red. **Below we summarize our updates to the manuscript.**

* We clarify that our model is not a neural model and the connection to neuroscience is at the functional level (Abstract, Intro, and Conclusion). In essence, our model is inspired by the hypothesis of "grandmother-cell", and we explore the utility of prototype cells as memory priors in a standard computer vision image generation task to seek to provide potential insights into the advantages of having these "grandmother" neurons in the visual cortex at a functional level.
* To avoid naming confusion, we rename the previously defined "prototype semantic cell" as "semantic cell" and the previously defined "prototype component cell" as "prototype cell".

* We add comparisons with other works that use prototype learning and memory bank in the related work and highlight the uniqueness of our work.
* We modify Figure 1 and Figure 2 to clarify the three projection heads $\theta(\cdot)$, $\phi(\cdot)$ and $\psi(\cdot)$ and how they are used in our model.
* Visualization results are added as new Figure 4 and Figure 5 in the main text to provide a further understanding of the semantic meanings of different clusters and prototypes formed in MoCA. Corresponding discussions are updated in Section 4.3.
* To verify that MoCA is not simply memorizing the training samples, we visualize the closest top 3 training data for the generated images in Appendix 8.8.
* We move the original analysis for MoCA's prototypes and some ablation study (original Figure 5, Figure 6 and subsection “Importance of memory organization”) to Appendix Sec.8.1 and Appendix 8.6.

---

### Decision · Program_Chairs · 2022-01-20

**Decision:**

Accept (Poster)

**Comment:**

This paper uses prototype memories for learning generative models. Inspired by the finding that there is sparse activity and complex selectivity in the supragranular layers of every cortical region, even primary visual cortex, the authors propose to use prototype memories at each level of the hierarchy, which marks their work as novel. They show superior performance in few shot image generation tasks.

The reviewers' scores were borderline (5,5,8), making this a case that required some AC consideration. The reviewers generally agreed that the paper was relevant and interesting, though the two more negative reviewers had some concerns about (1) the tests used, (2) the interpretation relative to neuroscience data, and (3) the novelty. After reading through the paper, the reviews, and the rebuttal's, the AC felt that the authors had made a decent attempt at addressing items (1) and (2), and item (3) was ultimately a subjective question. The authors were reasonably clear about what marks their work as novel, and it is certainly not *exactly* the same as previous work. Altogether, given these considerations, the AC felt that this paper deserved to be accepted, given the reasonable attempts from the authors to respond to the reviewers' concerns and an average score above acceptance threshold (though the scores did not change post-rebuttal, it should be noted).